# NEURAL DIFFUSION PROCESSES

## ABSTRACT

Gaussian processes provide an elegant framework for specifying prior and posterior distributions over functions. They are, however, also computationally expensive, and limited by the expressivity of their covariance function. We propose Neural Diffusion Processes (NDPs), a novel approach based upon diffusion models, that learns to sample from distributions over functions. Using a novel attention block we are able to incorporate properties of stochastic processes, such as exchangeability, directly into the NDP's architecture. We empirically show that NDPs are able to capture functional distributions that are close to the true Bayesian posterior. This enables a variety of downstream tasks, including hyperparameter marginalisation, non-Gaussian posteriors and global optimisation.

## 1 INTRODUCTION

Gaussian processes (GPs) offer a powerful framework for defining distributions over functions [26]. It is an appealing framework because Bayes rule allows one to reason consistently about the predictive distribution, allowing the model to be data efficient. However, for many problems GPs are not an appropriate prior. Consider, for example, a function that has a discontinuity at some unknown location. This cannot be expressed in terms of a GP, because it is impossible to express such behaviour by the first two moments of a multivariate normal distribution [23].

One popular approach to these problems is to abandon GPs, in favour of Neural network (NN) based generative models. Successful methods include the meta-learning approaches of Neural Processes (NPs) [8; 12; 2; 21], and VAE-based models [22; 6]. By leveraging a large number of small datasets during training, they are able to transfer knowledge across datasets at prediction time. Using NNs is appealing since most of the computational effort is expended during the training process, while the task of prediction usually becomes more straightforward. A further major advantage of a NN-based approach is that they are not restricted by the Gaussian assumption.

We seek to improve upon these methods by extending an existing state-of-the-art NN-based generative model. In terms of sample quality, the so-called probabilistic denoising diffusion model [31; 32; 10] has recently been shown to outperform existing methods on tasks such as image [24; 25], molecular structure [40; 11], point cloud [20] and audio signal [14] generation. However, the Bayesian inference of functions poses a fundamentally different challenge, one which has not been tackled previously by diffusion models.

**Contributions** We propose a novel model, the Neural Diffusion Process (NDP), which extends the use case of diffusion models to Stochastic Processes (SPs) and is able to describe a rich distribution over functions. NDPs generalise diffusion models to infinite-dimensional function spaces by allowing the indexing of random variables onto which the model diffuses. We take particular care to enforce known symmetries and properties of SPs, including exchangeability, and marginal consistency into the model, facilitating the training process. These properties are enforced with the help of a novel attention block, namely the *bi-dimensional* attention block, which guarantees equivariance over the ordering of (1) the input dimensionality and (2) the sequence (i.e., datapoints). From the experiments we draw the following two conclusions: firstly, NDPs are a clear improvement over existing NN-based generative models for functions such as Neural Processes (NPs). Secondly, NDPs are an attractive alternative to GPs for specifying appropriate (i.e., non-Gaussian) priors over functions. Finally, we present a novel global optimisation method using NDPs.

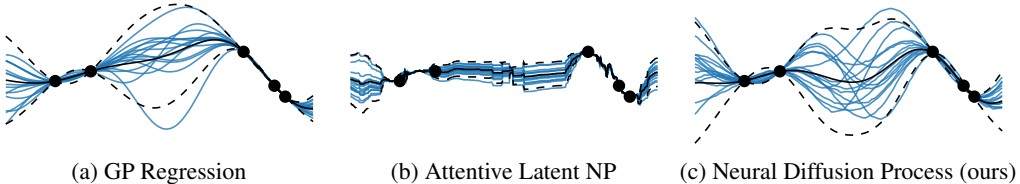

(a) GP Regression        (b) Attentive Latent NP        (c) Neural Diffusion Process (ours)

Figure 1: Conditional samples: The blue curves are posterior samples conditioned on the context dataset (black dots) from different probabilistic models.

## 2 BACKGROUND

The aim of this section is to provide an overview of the key concepts used throughout the manuscript.

### 2.1 GAUSSIAN PROCESSES

A Gaussian Process (GP) $f : \mathbb{R}^D \to \mathbb{R}$ is a stochastic process such that, for any finite collection of points $x_1, ..., x_n \in \mathbb{R}^D$ the random vector $(f_1, \ldots, f_n)$ with $f_i = f(x_i)$, follows a multivariate normal distribution [26]. In the case of regression, GPs offer an analytically tractable Bayesian posterior which provides accurate estimations of uncertainty —as shown in Fig. 1a where we observe that the predictive variance (dashed black line) elegantly shrinks in the presence of data (black dots).

GPs satisfy the Kolmogorov Extension Theorem (KET) which states that all finite-dimensional marginal distributions $p$ are consistent with each other under permutation (exchangeability) and marginalisation. Let $\pi$ be the a permutation of $\{1, \ldots, n\}$, then the following holds for the GP's joint:

$$p(f_1, \ldots, f_n) = p(f_{\pi(1)}, \ldots, f_{\pi(n)}) \quad \text{and} \quad p(f_1) = \int p(f_1, f_2, \ldots, f_n) \, \mathrm{d}f_2 \ldots \mathrm{d}f_n. \quad (1)$$

Despite their favourable properties, GPs are plagued by a couple of limitations. Firstly, encoding prior assumptions through analytical covariance functions can be extremely difficult, especially for higher dimensions [39; 29; 17]. Secondly, by definition, GPs assume a multivariate *Gaussian* distribution for each finite collect of predictions —limiting the set of functions it can model [23; 36; 38; 13; 5; 27]. We will revisit these limitations in the context of our experiments.

### 2.2 NEURAL PROCESSES AND THE META-LEARNING OF FUNCTIONS

Neural Process (NP) models [7; 8] have recently been introduced as an alternative to GPs —addressing the aforementioned problems. NPs utilise deep neural networks to define a rich probability distribution over functions. During training, NPs leverage a large number of small datasets, so that knowledge can be transferred across datasets during inference.

Despite being a promising research direction, NPs are severely limited by the fact that they do not produce consistent samples out-of-the-box. Broadly speaking, NPs have dealt with consistency in two ways: (1) by introducing an additional latent variable per function draw [8; 12], (2) by only modelling the marginals [7]. The former leads to consistent samples, but the likelihood for these models is not analytically tractable which requires crude approximations and ultimately limits their performance [21]. We observe this behaviour in Fig. 1b. The latter does not allow for sampling coherent functions at all, as no covariance information is available and all function values are modelled independently. We briefly summarise the family of NP models in Appendix D.4.

### 2.3 DIFFUSION MODELS

Our method relies on denoising diffusion probabilistic models (DPMs) [31], which we briefly summarise here. Diffusion models depend on two procedures: a *forward* and a *reverse* process, as illustrated in Fig. 2. The forward process consists of a Markov chain, which incrementally adds random noise to the data. The reverse process is tasked with inverting this chain in order to construct desired data samples from random noise alone.

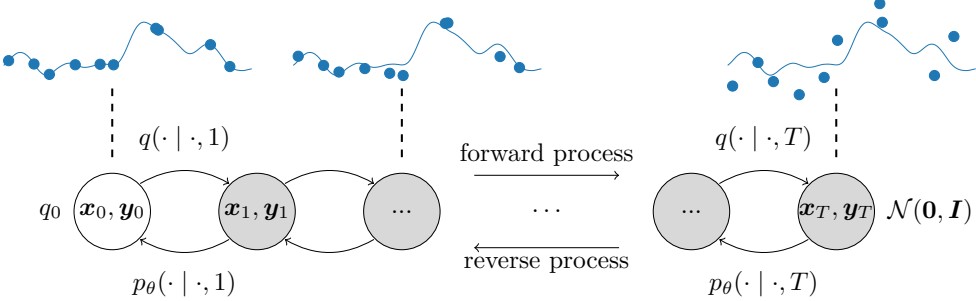

Figure 2: Illustration of the forward and reverse process of NDPs. (Left-to-right) During the forward process, function values (blue dots) from a function draw (blue line) are perturbed by increasing levels of noise until the points are given by white noise. (Right-to-left) The reverse process consists of gradually removing noise until we arrive at a data point.

The *forward process* starts from the data distribution $q(\boldsymbol{s}_0)$ and iteratively corrupts samples by adding small amounts of noise, which leads to a fixed Markov chain $q(\boldsymbol{s}_{0:T}) = q(\boldsymbol{s}_0) \prod_{t=1}^{T} q(\boldsymbol{s}_t \mid \boldsymbol{s}_{t-1})$ for a total of $T$ steps. The corrupting distribution is Gaussian $q(\boldsymbol{s}_t \mid \boldsymbol{s}_{t-1}) = \mathcal{N}(\boldsymbol{s}_t; \sqrt{1 - \beta_t}\boldsymbol{s}_{t-1}, \beta_t\mathbf{I})$. The magnitude of the noise in each step is controlled by a pre-specified variance schedule $\{\beta_t \in (0, 1)\}_{t=1}^{T}$. Note that there is no learning involved in the forward process, it simply creates a sequence of random variables $\{\boldsymbol{s}_t\}_{t=0}^{T}$ which progressively look more like white noise $q(\boldsymbol{s}_T) \approx \mathcal{N}(\mathbf{0}, \mathbf{I})$.

While the forward process is designed to be Markovian, the true reverse probability $q(\boldsymbol{s}_{t-1} \mid \boldsymbol{s}_t)$ requires the entire sequence. Therefore, the *reverse process* learns to approximate these conditional probabilities in order to carry out the reverse diffusion process. The approximation relies on the key observation that the reverse conditional probability is tractable when conditioned on the initial state $\boldsymbol{s}_0$: $q(\boldsymbol{s}_{t-1} \mid \boldsymbol{s}_0, \boldsymbol{s}_t) = \mathcal{N}(\boldsymbol{s}_{t-1}; \tilde{\mu}(\boldsymbol{s}_0, \boldsymbol{s}_t), \tilde{\beta}_t\mathbf{I})$. In this work, we follow Ho et al. [10] which approximate $\tilde{\mu}$ by estimating the initial state $\boldsymbol{s}_0$ from $\boldsymbol{s}_t$ and $t$. They propose the following paramterisation $\mu_\theta(\boldsymbol{s}_t, t) = \frac{1}{\sqrt{\alpha_t}}\big(\boldsymbol{s}_t - \frac{\beta_t}{\sqrt{1-\bar{\alpha}_t}}\boldsymbol{\epsilon}_\theta(\boldsymbol{s}_t, t)\big)$, $\alpha_t = 1 - \beta_t$, $\bar{\alpha}_t = \prod_{j=1}^{t} \alpha_j$ for $\tilde{\mu}$, and derive $\tilde{\beta}_t$ in closed-form (given in Appendix A). We refer to $\boldsymbol{\epsilon}_\theta$ as the noise model and implemented it as a neural network. Its architecture is detailed in Sec. 4.

The parameters $\theta$ of the noise model are optimised by minimising the objective $\mathbb{E}_{t,\boldsymbol{s}_0,\boldsymbol{\varepsilon}}[\boldsymbol{\varepsilon} - \boldsymbol{\epsilon}_\theta(\boldsymbol{s}_t, t)]$, where the expectation is taken over time $t \sim \mathcal{U}(\{1, 2, \ldots, T\})$, data points $\boldsymbol{s}_0 \sim q(\boldsymbol{s}_0)$, and a corrupted sample $\boldsymbol{s}_t = \sqrt{\bar{\alpha}_t}\boldsymbol{s}_0 + \sqrt{1 - \bar{\alpha}_t}\boldsymbol{\varepsilon}$ created from noise $\boldsymbol{\varepsilon} \sim \mathcal{N}(\mathbf{0}, \mathbf{I})$. Once the network is trained, generating samples from $q(\boldsymbol{s}_0)$ is done by simulating the reverse process. That is, starting from $\boldsymbol{s}_T \sim \mathcal{N}(\mathbf{0}, \mathbf{I})$ and iteratively denoising samples using $\boldsymbol{s}_{t-1} \sim \mathcal{N}(\boldsymbol{s}_{t-1}; \mu_\theta(\boldsymbol{s}_t, t), \tilde{\beta}_t\mathbf{I})$.

## 3 DIFFUSION PROBABILISTIC MODELS FOR STOCHASTIC PROCESSES

In this section we describe Neural Diffusion Processes (NDPs): a diffusion model designed for stochastic processes. We focus our attention on the difference between NDPs and traditional diffusion models used for images [24; 25], audio [14] or molecules [40; 11]. We defer the exposition of the NDP's noise model architecture to Sec. 4, where we also prove that NDPs satisfy the exchangeability and consistency conditions of KET.

### 3.1 FORWARD AND REVERSE PROCESS

**Data distribution** In NDPs, we are interested in sampling over the space of functions. A function, $f : \mathbb{R}^D \to \mathbb{R}$, is a fundamentally different random variable than, for example, an image. In an image, the pixel values are arranged on a predefined grid (height × width) and have an implicit ordering (e.g., left-to-right, top-to-bottom). Function values, in contrast, have no ordering and do not live on a predefined grid. Therefore, for NDPs to be useful, it is important that one can evaluate the generated samples everywhere in their input domain, i.e. $\forall x \in \mathbb{R}^D$.

To accommodate for this, we define the following data generating process for our training data: (1) sample the input dimensionality $D \sim \mathcal{U}(\{1, 2, \ldots, D_{\max}\})$ such that the domain is $\mathbb{R}^D$, (2) sample the number of point in the dataset $N \sim \mathcal{U}(\{1, 2, \ldots, N_{\max}\})$, (3) sample the $N$ points i.i.d. from a uniform distribution in $\mathbb{R}^D$ and collect them in $\boldsymbol{x}_0 \in \mathbb{R}^{N \times D}$ (we make the assumption that function inputs originate from a uniform measure over the space), (4) finally, set $\boldsymbol{y}_0 = f(\boldsymbol{x}_0)$, where $f$ can be a GP (i.e., $f \sim \mathcal{GP}(0, k_\psi)$ with $k_\psi$ the covariance function) or any other function class on which one wants to train (e.g., step functions as in Sec. 5.1). We obtain two correlated random variables $\boldsymbol{x}_0$ and $\boldsymbol{y}_0$ that are, respectively, the function's inputs and outputs.

**Diffusion** In the forward process, NDPs gradually add noise using a factorised distribution

$$q\left(\begin{bmatrix}\boldsymbol{x}_t\\\boldsymbol{y}_t\end{bmatrix} \mid \begin{bmatrix}\boldsymbol{x}_{t-1}\\\boldsymbol{y}_{t-1}\end{bmatrix}\right) = \mathcal{N}\left(\boldsymbol{y}_t; \sqrt{1-\beta_t}\,\boldsymbol{y}_{t-1}, \beta_t \mathbf{I}\right)\delta\left(\boldsymbol{x}_t - \boldsymbol{x}_{t-1}\right). \tag{2}$$

As shown in Fig. 2, this diffusion process corresponds to gradually adding Gaussian noise to the function values $\boldsymbol{y}_t$ while keeping the input locations $\boldsymbol{x}_t$ fixed across time. At $t = T$ the function values $\boldsymbol{y}_t$ should be indistinguishable to samples from $\mathcal{N}(\mathbf{0}, \mathbf{I})$, while $\boldsymbol{x}_T = \boldsymbol{x}_{T-1} = \ldots = \boldsymbol{x}_0$. In Sec. 5.3, we discuss a generalisation of this scheme where we perturb both the function inputs and outputs to obtain a diffusion model that can sample the joint $p(\boldsymbol{x}_0, \boldsymbol{y}_0)$. For now we are only interested in the conditional $p(\boldsymbol{y}_0 \mid \boldsymbol{x}_0)$ as this is the key quantity of interest in regression, which is why we keep the inputs fixed across time.

**Backward kernel** We parameterise the backward Markov kernel using a neural network that learns to de-noise the corrupted function values $\boldsymbol{y}_t$. In contrast, the input locations $\boldsymbol{x}_t$ have not been corrupted in the forward process which makes reversing their chain trivial. This leads to a parameterised backward kernel $p_\theta$ of the form

$$p_\theta\left(\begin{bmatrix}\boldsymbol{x}_{t-1}\\\boldsymbol{y}_{t-1}\end{bmatrix} \mid \begin{bmatrix}\boldsymbol{x}_t\\\boldsymbol{y}_t\end{bmatrix}\right) = \mathcal{N}\left(\boldsymbol{y}_{t-1}; \frac{1}{\sqrt{\alpha_t}}\left(\boldsymbol{y}_t - \frac{\beta_t}{\sqrt{1-\bar{\alpha}_t}}\boldsymbol{\epsilon}_\theta(\boldsymbol{x}_t, \boldsymbol{y}_t, t)\right), \tilde{\beta}_t \mathbf{I}\right)\delta\left(\boldsymbol{x}_t - \boldsymbol{x}_{t-1}\right). \tag{3}$$

In NDPs, the neural network model of the noise $\boldsymbol{\epsilon}_\theta : \mathbb{R}^{N \times D} \times \mathbb{R}^N \times \mathbb{R} \to \mathbb{R}^N$ has as inputs the function inputs $\boldsymbol{x}_t$, the corrupted function values $\boldsymbol{y}_t$, and time $t$. The network is tasked with predicting the noise that was added to $\boldsymbol{y}_0$ to obtain $\boldsymbol{y}_t$. The design of the network is specific to the task of modelling SP's and as such differs from current approaches. In the next section we discuss its particular architecture, but for now we want to stress that it is critical for the noise model to have access to the input locations $\boldsymbol{x}_t$ to make such predictions.

**Objective** Following Ho et al. [10], but substituting the NDP's forward and backward transition densities (Eqs. (2) and (3), resp.) leads to the objective (derived in full detail in Appendix A):

$$\mathcal{L}_\theta = \mathbb{E}_{t, \boldsymbol{x}_0, \boldsymbol{y}_0, \boldsymbol{\varepsilon}}\left[\left\|\boldsymbol{\varepsilon} - \boldsymbol{\epsilon}_\theta\left(\boldsymbol{x}_0, \sqrt{\bar{\alpha}_t}\boldsymbol{y}_0 + \sqrt{1-\bar{\alpha}_t}\boldsymbol{\varepsilon}, t\right)\right\|^2\right], \tag{4}$$

where we made use of the fact that $\boldsymbol{x}_t$ equals $\boldsymbol{x}_0$ for all timesteps $t$, and $\boldsymbol{y}_t = \sqrt{\bar{\alpha}_t}\boldsymbol{y}_0 + \sqrt{1-\bar{\alpha}_t}\boldsymbol{\varepsilon}$.

### 3.2 Prior and Conditional Sampling

**Prior** Once the network is trained, one can obtain prior function draws from the NDP at a specific set of input locations $\boldsymbol{x}_0$ by simulating the reverse process. That is, starting from $\boldsymbol{y}_T \sim \mathcal{N}(\mathbf{0}, \mathbf{I})$ and iteratively denoising samples using $p_\theta$ (Eq. (3)) for time $t = T, \ldots, 1$. This procedure leads to the samples in the left panes of Figs. 4 and 5.

**Conditional** NDPs are also able to draw conditional samples from the posterior $p(\boldsymbol{y}_0, \mid \boldsymbol{x}_0, \mathcal{D})$ where $\mathcal{D} = (\boldsymbol{x}_0^c \in \mathbb{R}^{M \times D}, \boldsymbol{y}_0^c \in \mathbb{R}^M)$ is the *context* dataset. Following recent advances in image in-painting [19], we adjust the reverse process to account for the information in the context dataset $\mathcal{D}$ as we want the conditional samples to be correlated and consistent with the context dataset. The conditional sampling process works as follows: We start by augmenting the intermediate state to contain the perturbed context dataset $\boldsymbol{y}_t^c$ and the de-noised state $\boldsymbol{y}_t$. This results in the augmented intermediate state $\breve{\boldsymbol{y}}_t = [\boldsymbol{y}_t^c, \boldsymbol{y}_t]$ (similarly, $\breve{\boldsymbol{x}}_0 = [\boldsymbol{x}_0^c, \boldsymbol{x}_0]$). The part of the augmented state $\breve{\boldsymbol{y}}_t$ that

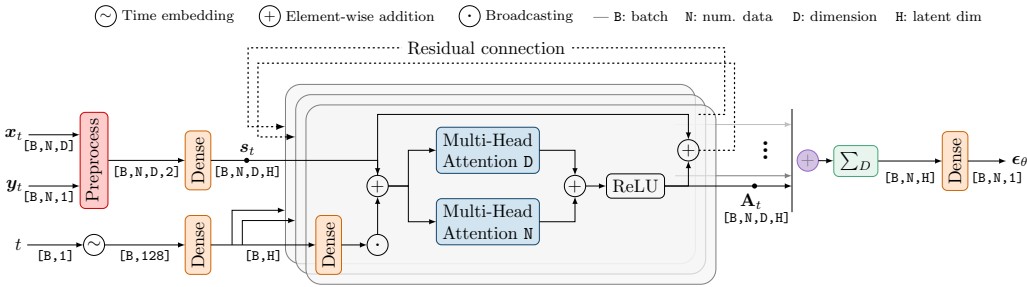

Figure 3: Architecture of the noise prediction model, utilised at each step within the Neural Diffusion Process. The greyed box represents the bi-dimensional attention block, as discussed in Section 4.2.

corresponds to the context points $\boldsymbol{y}_t^c$ is sampled using the forward process (left-hand-side), while $\boldsymbol{y}_t$ is sampled using the backward kernel (right-hand-side):

$$\boldsymbol{y}_t^c \sim \mathcal{N}\Big(\sqrt{\bar{\alpha}_t}\boldsymbol{y}_0^c, (1-\bar{\alpha}_t)\,\mathbf{I}\Big), \quad \text{and} \quad \boldsymbol{y}_{t-1} \sim \mathcal{N}\Big(\frac{1}{\sqrt{\alpha_t}}\big(\boldsymbol{y}_t - \frac{\beta_t}{\sqrt{1-\bar{\alpha}_t}}\epsilon_\theta(\check{\boldsymbol{x}}_0, \check{\boldsymbol{y}}_t, t)\big), \tilde{\beta}_t\mathbf{I}\Big) \quad (5)$$

Simulating this scheme from $t = T, \ldots, 1$ ensures that for each backward step (rhs of Eq. (5)) we leverage the context dataset as well as the randomness of the initial draw $\boldsymbol{y}_T \sim \mathcal{N}(\mathbf{0}, \mathbf{I})$. We show conditional samples using this scheme in Figs. 1c, 4 and 5, and list pseudocode in Appendix B.2.

# 4 ARCHITECTURE OF THE NOISE MODEL

NDPs implement the noise model $\epsilon_\theta$ as a neural network. In this section, we review its architecture and key components. In principle, *any* neural network could be used. However, if we wish for NDPs to mimic SPs, the noise model must learn to generate a prior distribution over functions. We expect such a prior to possess several key symmetries and properties, which will heavily influence our choice of architecture. We refer to Fig. 3 for an overview.

## 4.1 INPUT SIZE AGNOSTICITY

Before addressing the neural network invariances and equivariances, we focus our attention to a key property of the network: the NDP network is agnostic to dataset size $N$ and dimension $D$. That is, the weights of the network do *not* depend on the size of the inputs (i.e. $N$ nor $D$). This has as important practical consequences that it is not required to train different NDPs for datasets with different size or dimensionality. This makes it possible to train only a single model that handles downstream tasks with different $N$ or $D$ values. To achieve this functionality, NDPs start by reshaping the inputs $(\boldsymbol{x}_t \in \mathbb{R}^{N \times D}, \boldsymbol{y}_t \in \mathbb{R}^N)$ to $\mathbb{R}^{N \times D \times 2}$ by replicating the $\boldsymbol{y}_t$ outputs $D$ times before concatenating them with $\boldsymbol{x}_t$. Pseudocode for this operation is given in Appendix E.

## 4.2 BI-DIMENSIONAL ATTENTION BLOCK

One of the most fundamental properties of SPs, as specified by the KET, is an *equivariance* to the ordering of inputs. As such, shuffling the order of data points in the context dataset $\mathcal{D}$ or the order at which we make predictions should not affect the probability of the data (i.e., the data is exchangeable). Secondly, we also expect an *invariance* in the ordering of the input dimensions. Consider, for example, a dataset consisting of two features, the weight and height of people. We would not expect the posterior function to be different if we would swap the order of the columns in the training data. This is an important invariance encoded in many GP kernels (e.g., Matérn, RBF) but often overlooked in the neural net literature [18].

We accommodate for both desiderata using our newly proposed *bi-dimensional attention block*. We denote this block by $\mathbf{A}_t : \mathbb{R}^{N \times D \times H} \to \mathbb{R}^{N \times D \times H}$ as it acts on the preprocessed inputs $(\boldsymbol{x}_t, \boldsymbol{y}_t)$. At its core, the block consists of two multi-head self-attention (MHSA) layers [35]. The MHSA layers act on different axes of the input: one attends to the input dimension axis $d$, while the other attends

across the dataset sequence axis $n$. The outputs of the two are subsequently summed and passed through a non-linearity. This process is repeated multiple times by feeding the output back into the next bi-dimensional attention block using residual connections. The $\ell^{\text{th}}$ block is defined as

$$\mathbf{A}_t^\ell(\boldsymbol{s}_t^{\ell-1}) = \mathbf{A}_t^{\ell-1} + \text{ReLU}\left(\text{MHSA}_d(\boldsymbol{s}_t^{\ell-1}) + \text{MHSA}_n(\boldsymbol{s}_t^{\ell-1})\right) \quad \text{for } \ell = \{1, \ldots, L\}, \quad (6)$$

with $\boldsymbol{s}_t^0 = \boldsymbol{s}_t$ (i.e. the output of the preprocessing component –see diagram) and $\mathbf{A}_t^0 = \mathbf{0}$. Kossen et al. [15] introduced a similar block which alternates between attention across datapoints and attributes, whereas our bi-dimensional attention block acts on both simultaneously.

As the MHSA layers act on different axes, the component $\mathbf{A}_t$ is simultaneously *equivariant* to the order of the dimensions and dataset sequence (i.e., shuffling the rows and/or columns of the input will shuffle the output in the same way).

**Proposition 1.** *Let $\Pi_N$ and $\Pi_D$ be the set of all permutations of indices $\{1, \ldots, N\}$ and $\{1, \ldots, D\}$, respectively. Let $\boldsymbol{s} \in \mathbb{R}^{N \times D \times H}$ and $(\pi_n \circ \boldsymbol{s}) \in \mathbb{R}^{N \times D \times H}$ denote a tensor where the ordering of indices in the FIRST dimension are given by $\pi_n \in \Pi_N$. Similarly, let $(\pi_d \circ \boldsymbol{s})$ denote a tensor where the ordering of indices in the SECOND dimension are given by $\pi_d \in \Pi_D$. Then, $\forall \pi_n, \pi_d \in \Pi_N \times \Pi_D$:*

$$\pi_d \circ \pi_n \circ \mathbf{A}_t(\boldsymbol{s}) = \mathbf{A}_t\left(\pi_d \circ \pi_n \circ \boldsymbol{s}\right). \quad (7)$$

*Proof.* Follows from the invariance of MHSA layers, and the commutativity of operators $\pi_d$ and $\pi_n$ as detailed in Appendix C.2. $\square$

The noise model's $\epsilon_\theta$ outputs are obtained by element-wise adding the different bi-dimensional attention layers (see purple '+' in Fig. 3). This is followed by a sum over the input dimension axis (D). These operations introduce an invariance over D, while crucially preserving the equivariance over the dataset ordering N.

**Proposition 2.** *Let $\pi_n$ and $\pi_d$ be defined as in Proposition 1, then $\epsilon_\theta$ satisfies*

$$\pi_n \circ \epsilon_\theta(\boldsymbol{x}_t, \boldsymbol{y}_t, t) = \epsilon_\theta(\pi_n \circ \pi_d \circ \boldsymbol{x}_t, \pi_n \circ \boldsymbol{y}_t, t). \quad (8)$$

*Proof.* The claim follows from Proposition 1 and Zaheer et al. [41] as shown in Appendix C.3. $\square$

Crucially, by directly encoding these properties into the noise model, the NDP produces a set of random variables $\{y_t^1, \ldots, y_t^n\}$ at each time step $t$ that are exchangeable.

**Consistency** The second KET condition, consistency, is achieved when marginalising over the conditional distribution of a random variable $y^0$, we recover its original distribution $p(y^0) = \int p(y^0|y^1)p(y^1)\,\mathrm{d}y^1$. When it comes to consistency, NDPs are unlike NPs, in which the predictive distribution is created directly by the neural network. In NDPs the conditioning step is *not* performed by the neural network. Instead, conditioning takes place within the inherently probabilistic framework of the diffusion model itself. NDPs therefore naturally satisfy consistency, irrespective of the architecture of the NN. We include a formal proof in Appendix C.4, which relies on the fact that NDPs gradually transform samples from a consistent distribution $\mathcal{N}(\mathbf{0}, \mathbf{I})$ into the target distribution.

## 5 EXPERIMENTS

In the experiments we want to answer the following two questions: (1) What do NDPs provide on top of GPs? (2) How do NDPs compare, in terms of performance and applicability, to NPs? We address these questions in Sec. 5.1 and Sec. 5.2. Finally, in Sec. 5.3, we highlight a novel approach for global optimisation of black-box functions. All experiments (except Sec. 5.3) share the same model architecture illustrated in Fig. 3 with five bi-dimensional attention blocks. Full model configurations and training times are given in Appendix D.1.

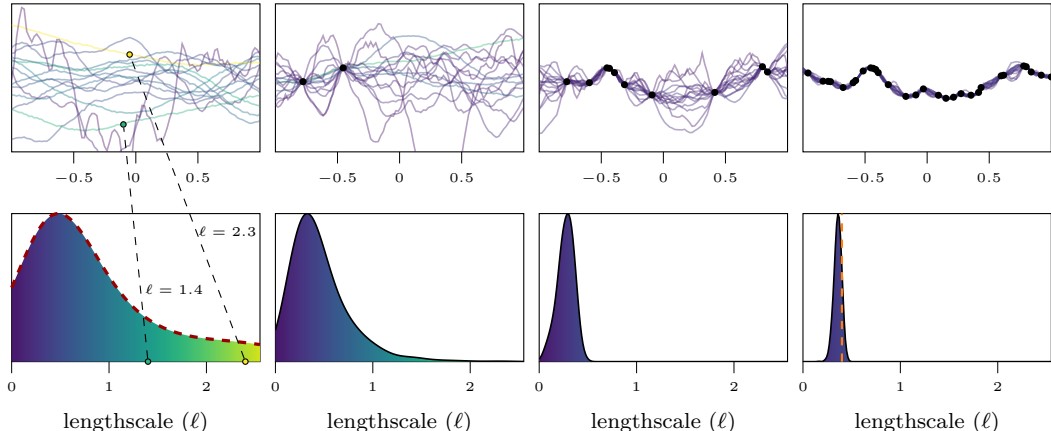

Figure 4: Kernel lengthscale marginalisation: Samples from the NDP, conditioned on an increasing number of data points (black dots), are illustrated in the top row. A sample is coloured according to its most likely lengthscale. As the NDP has no notion of a lengthscale, we infer the most likely lengthscale by retrospectively fitting a set of GPs and matching the lengthscale of the GP corresponding to the highest marginal likelihood to the sample. The bottom row shows a histogram of likely lengthscales from the produced samples. As more data points are provided, the distribution of likely lengthscales converges from the prior over lengthscales (red dashed line) to the lengthscale that was used to produce the data (orange dashed line).

## 5.1 NEURAL DIFFUSION PROCESSES VS. GAUSSIAN PROCESSES

GP regression models have three major limitations: cubic scaling of complexity $\mathcal{O}(N^3)$ with dataset size $N$, difficulty of choosing the right analytical prior covariance by hand, and the constraints associated with Gaussianity. NDPs provide a remedy to all three of these issues. Firstly, performing inference on a new dataset for a NDP is as costly as evaluating the noise model multiple times during the backward process. The complexity of this operation is $\mathcal{O}(TN^2)$, where $T$ is the number of timesteps. However, note that there is no training required to perform Bayesian inference on a new dataset after the initial training. This contrasts with the $\mathcal{O}(N^3)$ for training a GP per dataset. We address the remaining two limitations in the experiments below.

**Automatic marginalisation of hyperparameters** Conventionally, GP models optimise a point estimate of the hyperparameters. However, it is well known that marginalising over the hyperparameters can lead to significant performance improvements, albeit at an extra computational cost [16; 30]. A key capability of the NDP is its ability to produce realistic conditional samples across the full gamut of data on which it was trained. It can therefore in effect marginalise over the different hyperparameters it was exposed to, or even different kernels. Figure 4 demonstrates how the NDP produces samples from a range of different lengthscales. Indeed, this model was trained on a richer dataset consisting of samples from a GP with a Matérn-$\frac{3}{2}$ kernel, with a variety of lengthscales sampled from $\log \mathcal{N}(\log(0.5), \sqrt{0.5})$. In the lower pane, we can see that the NDP was able to capture this distribution, and that it narrows down on the true lengthscale as more data is observed.

**Non-Gaussian Posteriors** Each finite set of predictions of a GP follows a multivariate normal distribution. While this allows for handy analytic manipulations, it is also a restrictive assumption. For example, it is impossible for a GP to represent the covariance arising from 1D step functions when the step occurs at a random location in its domain [23]. For Fig. 5 we trained a NDP on exactly this prior where the domain is set to $[-1, 1]$. We observe that the NDP is able to correctly sample from the prior in (a), as well as from the conditional when we introduce datapoints in (b). In (c) we show the NDP's posterior of a marginal, which correctly shows the bimodal behaviour. The key takeaway from this experiment is that the NDP can infer a data-driven covariance, which need not be Gaussian and thus impossible for GPs to model.

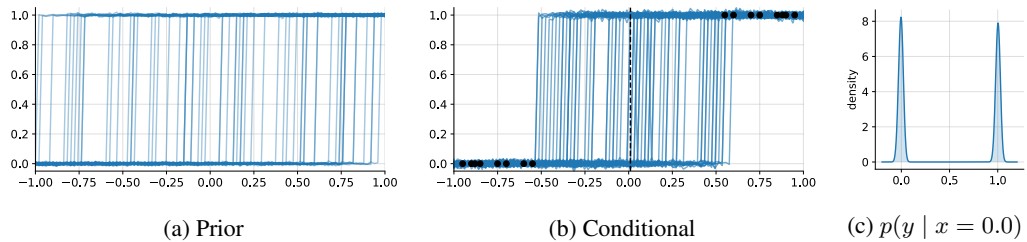

(a) Prior               (b) Conditional           (c) $p(y \mid x = 0.0)$

Figure 5: Representing a step function using NDP.

Table 1: Average Negative Log-Likelihood (NLL) across different synthetic datasets and input dimensions ($D_x$). The performance is measured on held-out functions that are conditioned on a random number of points between $1$ and ($30\,D_x$). Smaller is better.

| | Squared Exponential | | | Matérn-$5/2$ | | |
| --- | --- | --- | --- | --- | --- | --- |
| | $D = 1$ | $D = 2$ | $D = 3$ | $D = 1$ | $D = 2$ | $D = 3$ |
| NDP (ours) | $\mathbf{-0.38}\pm_{0.05}$ | $\mathbf{1.01}\pm_{0.03}$ | $\mathbf{1.20}\pm_{0.01}$ | $\mathbf{0.13}\pm_{0.05}$ | $\mathbf{1.15}\pm_{0.02}$ | $\mathbf{1.19}\pm_{0.01}$ |
| ANP [12] | $0.29\pm_{0.10}$ | $1.05\pm_{0.06}$ | $1.25\pm_{0.03}$ | $0.60\pm_{0.07}$ | $\mathbf{1.14}\pm_{0.05}$ | $1.29\pm_{0.02}$ |
| NP [8] | $0.67\pm_{0.06}$ | $1.23\pm_{0.04}$ | $1.35\pm_{0.02}$ | $0.84\pm_{0.04}$ | $1.26\pm_{0.03}$ | $1.36\pm_{0.01}$ |
| CNP [7] | $0.77\pm_{0.09}$ | $1.26\pm_{0.05}$ | $1.35\pm_{0.02}$ | $0.91\pm_{0.07}$ | $1.30\pm_{0.04}$ | $1.37\pm_{0.02}$ |
| trivial | $1.41\pm_{0.03}$ | $1.42\pm_{0.02}$ | $1.45\pm_{0.02}$ | $1.43\pm_{0.02}$ | $1.43\pm_{0.02}$ | $1.45\pm_{0.02}$ |

## 5.2 COMPARISON TO NEURAL PROCESSES

Neural Processes (NPs) [7; 8] are a family of models which are closely related to NDPs. NPs use a neural network to meta-learn a map from context datasets to predictive distribution over a target set.

**Empirical Evaluation** In Table 1, we empirically evaluate NDPs and NP models on synthetic datasets created using a Squared Exponential and a Matérn-$5/2$ kernel in different dimensions $D$. We compare NDP to Attentive NPs [12], Latent NP [8], Conditional NPs [7][1], and a trivial model which uses the empirical mean and variance as predictive distribution.

NDPs are superior to NPs in multiple areas. Firstly, from Table 1 we notice that NDPs outperform the NP family in terms of predictive performance. Secondly, NDPs produce consistent samples by design. For conditional NPs this is not the case —they are only able to predict the marginals. (Attentive) Latent NPs solve this problem by using a latent variable but due to crude approximations in the approximate inference scheme, they typically suffer from underfitting to the context set, leading to overestimated uncertainties, and samples that fail to pass through the context points [21; 22]. We observe this behaviour in our experiment, as shown in Figure 10 of the supplementary.

**Bayesian Optimisation** In this experiment we highlight that NDPs can be used in problems with $D \geq 3$. We tackle 3D and 6D minimisation problems using discrete Thompson sampling BO [28; 4; 33]. In this setting, the probabilistic model is used as surrogate of an expensive black-box objective. At each iteration the surrogate is evaluated at 128 random locations in the input domain and the input corresponding to the minimum value is selected as the next query point. The objective is evaluated at this location and added to the context dataset. Figure 6 shows the regret (distance from the true minimum) for the different models. We observe that the NDP almost matches the performance of GPR, which is the gold standard model on this type of task. NDPs also outperforms the NPs on both tasks. The important difference between GPR and the NDP is that the NDP requires *no training* during the BO loop, whereas GPR is retrained at every step.

## 5.3 NEURAL DIFFUSION PROCESSES TO MODEL THE JOINT $p(x, y)$

So far, we have used NDPs to model $p(\boldsymbol{y} \mid \boldsymbol{x}, \mathcal{D})$. This is a natural choice as in regression it is typically the only quantity of interest to make predictions. However, we can straightforwardly extend

---
[1]We use https://github.com/wesselb/neuralprocesses for all NPs as it provides SOTA implementations.

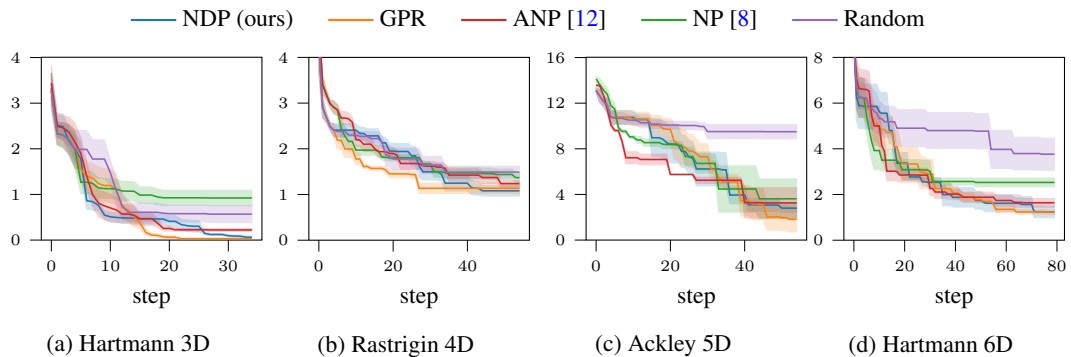

Figure 6: The regret of several probabilistic models used in Thompson sampling based Bayesian Optimisation on Hartmann 3D, Rastrigin 4D, Ackley 5D and Hartmann 6D optimisation problems.

NDPs to model the joint $p(\boldsymbol{x}, \boldsymbol{y} \mid \mathcal{D})$. For this, during the forward process we corrupt both the function inputs and outputs with additive Gaussian noise. The task of the reverse process now consists of denoising both the corrupted inputs $\boldsymbol{x}_t$ and outputs $\boldsymbol{y}_t$, which leads to the objective

$$\mathcal{L}_\theta = \mathbb{E}_{t, \boldsymbol{x}_0, \boldsymbol{y}_0, \boldsymbol{\varepsilon}_x, \boldsymbol{\varepsilon}_y} \left[ \|\boldsymbol{\varepsilon}_x - \boldsymbol{\epsilon}_\theta^x(\boldsymbol{x}_t, \boldsymbol{y}_t, t)\|^2 + \|\boldsymbol{\varepsilon}_y - \boldsymbol{\epsilon}_\theta^y(\boldsymbol{x}_t, \boldsymbol{y}_t, t)\|^2 \right], \quad (9)$$

where we highlighted the differences with Eq. (4) in orange. Importantly, in this case we require a noise model for both the inputs and outputs: $\boldsymbol{\epsilon}_\theta^x$ and $\boldsymbol{\epsilon}_\theta^y$, resp. We detail the architecture of this NN in the supplementary (Fig. 7). We design it such that $\boldsymbol{\epsilon}_\theta^x$ and $\boldsymbol{\epsilon}_\theta^y$ share most of the weights. The inputs to the NN are now the corrupted inputs $\boldsymbol{x}_t = \sqrt{\bar{\alpha}_t}\boldsymbol{x}_0 + \sqrt{1 - \bar{\alpha}_t}\boldsymbol{\varepsilon}_x$ and outputs $\boldsymbol{y}_t = \sqrt{\bar{\alpha}_t}\boldsymbol{y}_0 + \sqrt{1 - \bar{\alpha}_t}\boldsymbol{\varepsilon}_y$. This is in contrast to the previous NDP in which $\boldsymbol{x}_t = \boldsymbol{x}_0$ for all $t$.

**Global optimisation** By taking advantage of the NDP's ability to model the full joint distribution $p(x^*, y^* \mid \mathcal{D})$ we could conceive of a new global optimisation strategy. Consider conditioning the NDP model on our current belief about the minimum value $y^*$. This allows us to obtain samples from $p(x^* \mid y^*)$ which provide information about where the minima lies in the input domain according to the model. In Fig. 9 we illustrate a global optimisation routine using this idea. In each step, we sample a target from $p(y^*)$, which describes our belief about the minima. For the experiment, we sample $y^*$ from a truncated-normal, where the mean corresponds to the minimum of the observed function values, the variance corresponds to the variance of the observations. The next query point is selected by sampling $p(x^* \mid y^*, \mathcal{D})$, thereby systematically seeking out the global minimum. This experiment showcases NDP's ability to model the complex interaction between inputs and outputs of function —a task on which it was not trained.

## 6 Conclusion

**Limitations** As with other diffusion models, we found that the sample quality from a NDP improves with the number of diffusion steps $T$. This does however lead to slower inference times relative to other architectures such as GANs. Techniques for accelerating the inference process could be incorporated to ameliorate this issue. The method proposed by Watson et al. [37] is of particular interest for NDPs as we have a principled and differentiable metric to assess sample quality, namely the corresponding GP's marginal likelihood. Secondly, as is common with neural networks we found that it is important for test input points to lie within the training range, as going beyond leads to poor performance. This issue is also well-known for NP models Gordon et al. [9].

We proposed Neural Diffusion Processes (NDPs), a denoising diffusion model approach for learning probabilities on function spaces, and generating prior and conditional samples of functions. NDPs generalise diffusion models to infinite-dimensional function spaces by allowing the indexing of random variables. We introduced a new neural network building block, namely the bi-dimensional attention block, which wires dimension and sequence equivariance into the neural network architecture such that it behaves like a stochastic process. We empirically show that NDPs are able to capture functional distributions that are richer than GPs and more accurate than NPs.

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
