# OpenReview forum: "Neural Diffusion Processes"
_ICLR.cc/2023/Conference — Submitted to ICLR 2023_

### Official Review · Reviewer_YxyB · 2022-10-16

**Confidence:** 2
**Correctness:** 3
**Technical Novelty And Significance:** 3
**Empirical Novelty And Significance:** 2
**Recommendation:** 6

**Clarity, Quality, Novelty And Reproducibility:**

**Clarity**: several parts of the paper was a bit vague to me and thus may not be very clearly presented
 - In Sec. 2.2, what is the math details behind Neural Processes, especially attentive neural processes? Considering that it generated quite comparable results to NDP in Table 1 and Figure 6, some discussions about its strengths, weakness and similarities to NDP would be beneficial.
- The bi-dimensional attention block seems to be a key contribution of the paper. A clear definition in the main paper about it as well as the entire noise prediction model would help with understanding. Now it everything is in the Appendix except the diagram in Figure 3.

**Quality**: the paper provided detailed derivations on the following aspects, making it a coherent work
- forward and backward computation
- the loss derivation and training algorithm
- prior and conditional sampling
- permutation equivariance and dimensionality invariance

**Novelty**: the idea of applying the diffusion model to distributions of functions is relatively new and not trivial

**Strength And Weaknesses:**

The submission mentioned the complexity of GP is $O(N^3)$ and NDP is $O(TN^2)$:
- How was $T$ selected in the experiments in practice? In many plots $N$ is small, how did $T$ compare to $N$?
- I assume the proposed method used auto diff and gradient updates to minimize the objective. How long did it take to converge? Compared to GP regression, which has closed-form updates, is the computation of NDP really faster?

I can see Sec. 5.3 also added noise to the input $x$ to model the joint distribution $p(x, y)$. Could you explain the difference between the Bayesian optimization in Sec. 5.2 and the global optimization in Sec. 5.3?



**Summary Of The Paper:**

The paper proposed Neural Diffusion Processes to specify distribution over functions. It uses a similar learning approach to diffusion models, but generalized the dimensions to infinities. Prior and conditional sampling from NDP were discussed. The authors designed a novel bi-dimensional attention block in the architecture of the noise model, to ensure dimension and sequence equivariance, and therefore making it satisfy the properties of stochastic processes. Experiments on synthetic datasets compared NDP with Gaussian Process and Neural process, and presented a way for global optimization of black-box functions.

**Summary Of The Review:**

The paper provides a new of using the diffusion model, ie to model the distribution of functions. This is not a trivial extension, and the paper provides rounded support to justify its correctness. Experiment results are well-rounded to show the proof of concept, but not very significant from the practical point of view. Writing could be improved to put more emphasis on the key parts of the contributions. Given these, I would give it a recommendation of marginally above the acceptance threshold.

---

> ### Author Response · Authors · 2022-11-11
> **Response**
>
> > Clarity: several parts of the paper was a bit vague to me and thus may not be very clearly presented
>
> Thank you for your comment, we have extended and improved our background and architecture sections, as well as our appendices to incorporate your feedback.
>
> > In Sec. 2.2, what is the math details behind Neural Processes, especially attentive neural processes? Considering that it generated quite comparable results to NDP in Table 1 and Figure 6, some discussions about its strengths, weakness and similarities to NDP would be beneficial.
>
> In short, Neural processes use an encoder and decoder architecture. The encoder is a neural network which operates on a dataset $(X,Y)$ to output a dataset representation $r = enc(X, Y)$. Using this representation, a decoder predicts the function output at a test location $f^* = dec(x^*, r)$. An attentive Neural processes (ANPs) uses an attention mechanism (Vaswani, 2017) to parameterise the encoder and decoder networks. While ANPs perform better than plain NPs, they suffer from two key weaknesses. Firstly, their predictions result in jittery functions as a result of shifting attention patterns. This behaviour is well-known and can also be observed in the top row of Fig. 10 in the supplementary. Secondly, ANP do not encode dimensionality invariance in their architecture, which we show is critical for tabular data applications.
>
>
> > The bi-dimensional attention block seems to be a key contribution of the paper. A clear definition in the main paper about it as well as the entire noise prediction model would help with understanding. Now it everything is in the Appendix except the diagram in Figure 3.
>
> We agree with the reviewer that we missed a definition of the attention block in the main paper. We have corrected this by changing the title of section 4.2 to "Bi-dimensional attention block" and added a clear mathematical description and a more detailed explanation of the component. Thank you for drawing our attention to this.
>
> > How was $T$ selected in the experiments in practice?
>
> In the experiments, $T$ was set to $500$. On a single GPU, inference on a new dataset takes roughly ~5 seconds. We added the dependence of $T$ on the sample quality in the limitation sections of the paper. Reducing the number of steps in the reverse process is an active area of research, and successful approaches can directly be translated to NDPs. The method proposed by Watson et al. (2021) is of particular interest as NDPs have a principled and differentiable metric to assess sample quality, namely the corresponding GP's marginal likelihood.
>
> > I assume the proposed method used auto diff and gradient updates to minimize the objective. How long did it take to converge? Compared to GP regression, which has closed-form updates, is the computation of NDP really faster?
>
> While the NDP's pretraining procedure does indeed make use of auto diff and gradient updates, which took roughly 15 minutes to converge, this is not a task which needs to be repeated for different datasets. Unlike GP regression, performing inference on a new dataset for a NDP requires only a single pass through the reverse process. There is no training required to perform Bayesian inference on a new dataset. This is in contrast to GPs which require optimisation for each new dataset.
>
> > I can see Sec. 5.3 also added noise to the input  to model the joint distribution . Could you explain the difference between the Bayesian optimization in Sec. 5.2 and the global optimization in Sec. 5.3?
>
> We note that a typo in the caption of Figure 7 (Figure 9 in the revised manuscript) incorrectly stated that in 5.3 we sample from $p(y|x, D)$, which gave a misleading impression that is it more similar to Bayesian Optimisation. We have now fixed this error and we thank the reviewer for drawing our attention to this. In 5.2, optimisation proceeds by drawing samples from the posterior $p(y|x, D)$ whereas in 5.3 optimisation proceeds by drawing samples from the posterior $p(x|y, D)$.
>
> To go into a little more detail, in Bayesian optimisation the model samples are used to evaluate an acquisition function and the next point is determined by the acquisition function value at different input locations. The global optimisation procedure outlined in Sec. 5.3 directly samples the input locations that are equal to a desired (low) y value. This provides a fundamentally different global optimisation procedure, that is not based on an acquisition rule.
>
> We thank the reviewer for their comments and hope that these clarifications could lead to an uplift in their score.

---

### Official Review · Reviewer_d5b3 · 2022-10-25

**Confidence:** 3
**Correctness:** 4
**Technical Novelty And Significance:** 4
**Empirical Novelty And Significance:** 3
**Recommendation:** 8

**Clarity, Quality, Novelty And Reproducibility:**

The paper is well written and organized. Experiments are provided in detail to enable reproducibility of the results. Empirical results are presented with error bars. To my knowledge, this is the first study using diffusion models for stochastic processes, so the contributions are quite unique.

**Strength And Weaknesses:**

Strengths:
- The paper makes a novel contribution combining diffusion modeling with stochastic processes.
- Gaussian processes’ cubic scaling with dataset size is a big limitation and this paper offers a technique reducing computational complexity to O(TN^2). With the recent advancements in diffusion modeling, T is reduced to as few as 4 steps (Salimans and Ho, 2022), which might be an important point for this technique’s adoption.
- The proposed technique also offers unique properties ranging from automatic marginalization of hyperparameters without added computational cost, and modeling the joint distribution p(x,y) enabling interesting global optimization schemes.

Weaknesses:
- The experiments are conducted using simulated data. It would be interesting to see the performance of the proposed technique in real data settings. As an example, neural processes (NP) (Garnelo et al. 2018) explore image completion as a regression task with application to MNIST and CelebA datasets. It would be interesting to see comparisons between the proposed approach and NP.

Other comments:
- Kossen et al. (2022) proposes non-parametric transformers where the architecture consists of layers of attention mechanism across datapoints and attributes to enable reasoning using an entire dataset. Can the authors comment on the similarities between this and the proposed bi-dimensional attention block?
- Why do you show q(x_t|x_{t-1}) and p_\theta(x_{t-1}|x_t) in Eqns 2 and 3 since x’s are treated deterministically?


Salimans, T. and Ho, J., 2022. Progressive distillation for fast sampling of diffusion models. arXiv preprint arXiv:2202.00512.

Garnelo, M., Schwarz, J., Rosenbaum, D., Viola, F., Rezende, D.J., Eslami, S.M. and Teh, Y.W., 2018. Neural processes. arXiv preprint arXiv:1807.01622.

Kossen, J., Band, N., Lyle, C., Gomez, A.N., Rainforth, T. and Gal, Y., 2021. Self-attention between datapoints: Going beyond individual input-output pairs in deep learning. Advances in Neural Information Processing Systems, 34, pp.28742-28756.


**Summary Of The Paper:**

This paper proposes to use a diffusion based approach to model stochastic processes and thus enabling sampling from distributions over functions. A novel attention mechanism is proposed to build exchangeability and marginal consistency into the neural network model architecture. The model is trained to capture prior predictive distributions and to obtain posteriors, conditioning information is incorporated at inference time. The proposed approach reduces the computational complexity with respect to Gaussian processes and enables non-Gaussian priors. Experiments demonstrate automatic hyperparameter marginalization properties and applications to Bayesian optimization.


**Summary Of The Review:**

Diffusion models are beating state of the art in various problems and this paper proposes their application to stochastic processes, which is a timely and novel contribution. The paper demonstrates the approach through interesting applications in hyperparameter marginalization and joint distribution modeling. It is a good paper, I recommend acceptance. The only reason why I am not recommending a strong acceptance is due to the lack of real data experiments.

---

> ### Author Response · Authors · 2022-11-11
> **Response**
>
> > The experiments are conducted using simulated data. It would be interesting to see the performance of the proposed technique in real data settings. As an example, neural processes (NP) (Garnelo et al. 2018) explore image completion as a regression task with application to MNIST and CelebA datasets. It would be interesting to see comparisons between the proposed approach and NP.
>
> Thank you for your comment. We have extended our Bayesian optimisation experiment (Figure 6) with two more benchmarks: Ackley and Rastrigin. We note that these experiments are so-called sim-2-real: we train on synthetic data (i.e. Gaussian process draws) but evaluate the real-world objective that was not seen during training. In this paper, we have explicitly refrained from working with image data given the extraordinary performance of diffusion models on images (e.g., DALLE2 and Imagen) when modelling them on a fixed grid rather than as functions as NDPs do. Moreover, our architecture's dimension invariance is key for tabular data in our experiments but is not a sensible prior for human readable images.
>
> > Kossen et al. (2022) propose non-parametric transformers where the architecture consists of layers of attention mechanism across data points and attributes to enable reasoning using an entire dataset. Can the authors comment on the similarities between this and the proposed bi-dimensional attention block?
>
> Thank you for the reference. We have included this in the paper (below eq. 6). The key difference between the neural net in Kossen et al. (2022) and our bi-dimensional attention block is that in Kossen et al. (2022) the network alternates between attention across datapoints and attributes, whereas our bi-dimensional attention block acts on both simultaneously.
>
> > Why do you show $q(x_t|x_{t-1})$ and $p_\theta(x_{t-1}|x_t)$ in Eqns 2 and 3 since x’s are treated deterministically?
>
> You are right, both formulations are correct. As you say, one can drop them from the equations and treat them deterministically. Alternatively, one can interpret the $x$'s as a random variable with mass at a single point. We chose the latter formulation as it sets the scene better for Section 5.3 on modelling the joint, where $x$ is not a point mass anymore.
>
> Thank you for your insightful review. We hope you can share your enthusiasm for the paper in the upcoming discussion period.

---

### Official Review · Reviewer_6nma · 2022-10-25

**Confidence:** 4
**Correctness:** 2
**Technical Novelty And Significance:** 2
**Empirical Novelty And Significance:** 2
**Recommendation:** 3

**Clarity, Quality, Novelty And Reproducibility:**

The paper is fairly easy to read, the topic is interesting, but the paper contains a few mistakes.

**Strength And Weaknesses:**

- **Strength:** Any attempt to mary the pragmatism, effectiveness and scalability of deep learning with the "principledness" of Bayesian nonparametrics is always laudable. This paper is no exception.

- **Weaknesses:**

**1- Incorrect Construction:** Why are the sample size and the input dimension drawn rather than fixed, let alone from a uniform distribution? As a stochastic process, what space are NDPs indexed in?

**2- Wrong Consistency Proof:** One would expect that what makes NDPs a generalization of diffusion models (DMs) is that the forward and reverse processes operate on functions, not simply fixed and finite dimensional inputs, through an arbitrary (but deterministic) number of inputs and function values.

In this case, consistency ought to be proven not across time, but with respect to the choice of inputs. For instance, you need to show that if you have 3 inputs $(a, b, c)$ and associated outputs $(y_a, y_b, y_c)$ at time $0$, whether you consider all 3 inputs, $a$ and $b$, $b$ and $c$, or only $b$, at time $T$ the $b$-marginals are all the same. The same goes for the reverse process.

**3- Insufficient Motivation:** How does the infinite dimensional extension of DMs improve on DMs?

**4- Incomplete Literature Review:** There is an extensive literature on expressive kernels (e.g. Sparse Spectrum Kernels, Spectral Mixture Kernels, Generalized Spectral Kernels (stationary and non-stationary), etc.) that is worth reviewing when comparing GPs with NDPs. Additionally, at the end of the day the forward process should map a structured process into a Gaussian white noise, and the reverse process should map a Gaussian white noise into a structured process. Many such approaches have already been proposed that would have been useful to review here (e.g. the Karhunen–Loève expansion, etc.).

**5- Other Mistakes:**  E.g. The section "Non-Gaussian Posteriors" contains several mistakes. First, the covariance function of any $L^2$ process is also the covariance function of a mean-zero GP. So one cannot say that "it is impossible for a GP to represent the covariance arising from 1D step functions when the step occurs at a random location in its domain". Similarly, the sentence "The key takeaway from this experiment is that the NDP can infer a data-driven covariance, which need not be Gaussian and thus impossible for GPs to model." is incorrect. There is no such a thing as a Gaussian covariance, and expressive covariance functions (e.g. GSKs) should have been considered here.







**Summary Of The Paper:**

The authors introduce a new class of stochastic processes that they think of as a generalization of diffusion models to infinite-dimensional inputs.


**Summary Of The Review:**

The attempted generalization of diffusion models to functions is interesting, but the work requires more polishing before it is ready for publication.

---

> ### Author Response · Authors · 2022-11-11
> **Response**
>
> > Incorrect Construction
>
> An important property of an NDP model is that the same model can handle any input dimension (D) and dataset size (N). We reflect this in our training data by sampling the size and the dimension of a dataset from a uniform distribution. In practice, we sample the training function inputs from $[-1, 1]^d$, which is the space in which the NDPs indexes. As is common with neural networks we found that it is important for test points to lie within the training range, as going beyond -1 or 1 leads to poor performance. This issue is also well-known for Neural process models (Gordon et al., 2019). We have added this to the limitation section of our paper, and thank the reviewer for drawing this to our attention.
>
> > Wrong Consistency Proof
>
> The proof in Section C.3 of the supplementary material covers exactly the consistency requirement that you describe. The time element is merely used as an aid to complete the proof. We start with a white noise distribution which trivially satisfies self-consistent marginals. We then show that this consistency is preserved over individual time steps. Thus it still holds at the end of the process.
>
>
> > Insufficient Motivation
>
> You are correct that this paper does not improve diffusion models in itself. However, this was not the goal of the paper. The primary goal of this work is to extend the use-case of diffusion models, which have been shown to work extremely well on many data modalities, to functions.
>
> We expect our work to be relevant to the research community for multiple reasons. Firstly, both diffusion models and generative models over functions are active areas of research. Secondly, we show a clear improvement upon the current state-of-the-art Neural processes model -- both theoretically (e.g., consistency) and empirically. Finally, given the wide adoption of NPs in applications (e.g., [1-3]) we expect our model to also be well-received by the wider probabilistic modelling community as we believe NDPs can be a drop-in replacement for the NP model and improve the performance.
>
>
> Going beyond supervised learning or generative modelling, we show a novel approach to learn the joint density $p(x, y)$. This is not possible with existing diffusion models and allows for interesting applications such as global optimisation, among others.
>
> [1] Conditional Neural Processes for climate downscaling in an environmental journal: https://gmd.copernicus.org/articles/15/251/2022/.
>
> [2] Conditional Neural Processes for chemistry: https://arxiv.org/pdf/2210.09211.pdf.
>
> [3] Learning fields with symmetries using CNPs: http://proceedings.mlr.press/v139/holderrieth21a/holderrieth21a.pdf.
>
>
> > Incomplete Literature Review
>
> Thank you for the references, we have extended the background section on Gaussian processes (Section 2.1) to include spectral kernels. However, we note that even with more complicated spectral kernels it is not possible to represent non-Gaussian behaviour in the latent process. We further illustrate this by replicating our experiment (Figure 5) with the spectral mixture kernel (Wilson et al., 2013). As shown in the figure, each marginal of the process will always be Gaussian distributed.
>
> - Spectral Mixture Kernel with Q = 1: https://pasteboard.co/CiZp1KGamWD1.png
> - Spectral Mixture Kernel with Q = 3: https://pasteboard.co/uCNDzl9uwJL7.png
>
> > Other Mistakes
>
> We respectfully disagree. By definition, a Gaussian Process is a stochastic process, such that every finite collection of random variables has a multivariate normal distribution. A function with a step at an unknown location leads to a bimodal distribution that cannot be captured by a GP -- irrespective of the choice of kernel. The example of a step function has also been studied by Neal (1998) in section 3 "Limitations of Gaussian processes". Above, we have demonstrated that even with more complicated spectral kernels we can not bypass the Gaussian nature of the GP.
>
> We thank the reviewer for their critical feedback. Based on your comments we improved our related work section, included a limitation section and ran additional experiments with spectral kernels. We sincerely hope that this could lead to an improvement in your assessment of the paper.
>
> **References**
>
> - R. M. Neal. Regression and classification using gaussian process priors. 1998.
>
> - Wilson et al. Gaussian Process Kernels for Pattern Discovery and Extrapolation. 2013.
>
> - Gordon et al., Convolutional conditional neural processes. ICLR, 2019.

---

> > ### Comment · Reviewer_6nma · 2022-11-21
> > **Post Response Comment**
> >
> > I thank the authors for their response.
> >
> > **Re: Incorrect Construction** One of the most basic properties of a stochastic process is that it ought to be indexed on some well-defined space and take value in some well-defined space. In this paper as well as in the authors response, it is unclear what space an NDP is defined on and what space an NDP takes value in. The authors can't both say "the same model can handle any input dimension (D) and dataset size (N)." and  also "we sample the training function inputs from $[-1, 1]^d$, which is the space in which the NDPs indexes".
> >
> > I stand by the fact that this process is incorrectly constructed.
> >
> > **Re: Wrong Consistency Proof** Not only is consistency dealt with in the Appendix in C4, not C3, but C4 **does not** prove consistency. If the authors want to consider NDP a stochastic process that generalizes diffusion models to functions, they need to ensure that their proposal yields mutually consistent finite dimensional distributions w.r.t the choice of function inputs/values they feed to their diffusion model, not just w.r.t. time. This has **not** been shown in this paper or its appendix. The authors cannot claim an NDP acts on functions unless this is done; NDP would simply be a generalization of diffusion models to [...] large dimensional inputs which can be interpreted as **a specific** set of function inputs/values.
> >
> > Speaking of inconsistency and lack of rigor, the authors seem to be confusing random variables and stochastic processes at times (e.g. on page 19 appendix C4, the authors claim that the Kolmogorov consistency of a random variable ($y_T$ is a Gaussian random variable) is "trivially satisfied".
> >
> > **Re: Insufficient Motivation** Unfortunately the extension to functions is not proved until the authors prove consistency w.r.t. the choice of function values, which they haven't. NDPs act on a specific set of function values. For the authors to claim NDPs act on functions, they need to show that when an NDP operate on any two distinct but overlapping set of functions inputs/values, the marginal distribution of the overlapping function inputs/values at time $T$ implied by both models is the same. Otherwise, an NDP would be specific to the choice of function inputs/values fed to it, not specific to the choice of functions. More importantly, it is unclear to me how this can be achieved while considering the dimensional of the domain of the function variable.
> >
> > **Re: Incomplete Literature Review** Kindly note that your review is incomplete without generalized spectral kernels, which are a plug-in replacement for spectral mixture kernels and include non-stationary kernels and spectral mixture kernels as a special case.
> >
> > **Re: Other Mistakes** I invite the authors to re-read my original comment as they seem to have misunderstood it. That a Gaussian Process cannot represent a random step function with unknown jump location is a fact I never disputed. What is however incorrect is to say that "it is impossible for a GP to represent the covariance arising from 1D step functions". A GP can model **any** covariance function! Again, the covariance function of any $L^2$ process is the covariance function of a mean-zero GP.
> >
> > What the authors seem to have missed and that is fundamental to understand here is that, if a GP cannot represent a random step function with unknown jump location, *it has nothing to do with its covariance function*! It has everything to do with higher moments!
> >
> > Similarly, there is no such a thing as a "Gaussian covariance" like the authors implied in the sentence "[...] the NDP can infer a data-driven covariance, which need not be Gaussian [...]". Again, any covariance function is the covariance function of a mean-zero GP!
> >
> > **Summary:** I stand by my original recommendation as, for a theoretical contribution, this paper has far too many flaws, some fatal. Specifically, without a correct proof of consistency NDPs do not diffuse functions, they diffuse a ***specific*** finite-dimensional set of function inputs/outputs which is no different than standard diffusion models, in a way that could very well be characteristic of the chosen set of function inputs/outputs.

---

> > > ### Author Response · Authors · 2022-11-23
> > > **Response**
> > >
> > > We thank the reviewer for engaging in the discussion.
> > >
> > > **Incorrect construction**
> > >
> > > We stand by the fact that this process is correctly constructed, and we fail to see any inconsistency in claiming that NDPs can handle arbitrary $D$ and $N$ while being confined to the unit hypercube. We also want to point to our experimental evaluation which demonstrates that this construction works in practice.
> > >
> > >
> > > **Insufficient Motivation and Consistency Proof**
> > >
> > > You are correct that during training we diffuse upon a finite set of inputs and corresponding function values. However, the score function we approximate is a neural network $\epsilon_\theta(x,y,t): \mathcal{X} \times \mathbb{R} \times \mathbb{R} \rightarrow \mathbb{R}$ which depends on $x \in \mathcal{X}$. This allows us at inference time to evaluate the samples at arbitrary locations in $\mathcal{X}$. In our case $\mathcal{X}$ is the hypercube $[-1, 1]^d$. This is why we emphasize that a NDP acts on *functions* because of its ability to evaluate the samples at arbitrary locations à posteriori is a capability that a standard diffusion model doesn't possess.
> > >
> > > Furthermore, a Neural Diffusion Process has the same downstream functionality as a Neural Process (NPs), but as we show, NDPs empirically outperform NPs on a range of tasks. As NPs have been well-received and are often used by the community we expect NDP to also be of interest to the ICLR audience.
> > >
> > > **Other Mistakes**
> > >
> > > We agree with the reviewer that the inability of a GP to represent a step function at an unknown location is due to characterising the GP by only its first two moments. We have discussed this in the first paragraph of the paper as we explicitly write: "[...] because it is impossible to express such behaviour by the first two moments of a multivariate normal distribution." We are happy to rephrase the term 'Gaussian covariance' to a more verbose form that leaves less room for ambiguity. However, we can reassure the reviewer that the term does 'exist', it refers to a covariance that receives no contributions from higher order moments (see for example the 2020 NeuIPS paper entitled "Robust Gaussian covariance estimation in nearly-matrix multiplication time"). Nonetheless, we hope the reviewer agrees that the main point of this experiment is to highlight a functionality of our NDP (i.e. the ability to model step functions at arbitrary locations) which a GP fails to do.
> > >
> > >
> > > **Incomplete Literature Review**
> > >
> > > We have included Generalized Spectral Kernels to the background section of the manuscript. We believe the reviewer agrees that a GP cannot represent a random step function with an unknown jump location. A GP with the suggested non-stationary GSK is no exception to this. We, therefore, see limited relatedness between this specific kernel and our "step function" experiment.

---

### Official Review · Reviewer_sTEa · 2022-10-29

**Confidence:** 4
**Correctness:** 4
**Technical Novelty And Significance:** 4
**Empirical Novelty And Significance:** 3
**Recommendation:** 6

**Clarity, Quality, Novelty And Reproducibility:**

**Novelty**: The proposed method theoretically generalizes diffusion models to function spaces, which seems quite interesting and looks novel. NDPs model the distribution of input-output pairs and address it with a specially-designed architecture of neural networks. The novelty seems to be the main strength of this paper.

**Quality**: The proposed method is technically sound, and the theoretical results are solid to demonstrate the strength of NDPs compared to previous NP models.

**Clarity**: The paper is well-written, and the figures are illustrative. Notations are clean.

**Reproducibility**: The paper seems to provide sufficient information in the appendix to reproduce the results although a code is not provided.

**Strength And Weaknesses:**

**Strength**
- The paper provides novel contributions (NDP formulation, bidirectional attention module) and properly enforces important properties of stochastic processes into the neural architecture. The model is invariant to the input dimension, which is a desired property for neural net families to model function spaces.

**Weakness**
- The empirical comparison seems interesting but the authors should consider a broader family of BO objectives beyond the Hartmann function, which is quite easy to imitate the inference of GP. How does NDP perform on the Rastrigin function or the Ackley function in high-dimensional space?

**Question**
- NDP provides an interesting idea to model the joint distribution of a given function space in Section 5.3. However, it is confusing to me since I think it is nothing more than changing the delta distribution in equations (2) and (3). Why do you separate those contents from Section 3?

**Summary Of The Paper:**

This paper proposes a diffusion model, Neural Diffusion Process (NDP), to represent a distribution over function spaces by parameterizing the score model with a neural network. To guarantee the essential properties of stochastic processes, e.g., exchangeability, the authors propose a bi-dimensional attention block, which ensures equivariance over the input dimensionality and sequence order.

**Summary Of The Review:**

I think the paper provides an interesting idea and experimental results (marginalization, Bayesian optimization) on synthetic data demonstrate several interesting features of the proposed method.

---

> ### Author Response · Authors · 2022-11-11
> **Response**
>
> > The empirical comparison seems interesting but the authors should consider a broader family of BO objectives beyond the Hartmann function, which is quite easy to imitate the inference of GP. How does NDP perform on the Rastrigin function or the Ackley function in high-dimensional space?
>
> As suggested, we have incorporated two additional benchmarks (Ackley and Rastrigin) into the Bayesian optimisation experiment in Figure 6. Over the four different optimisation tasks, we see that a Neural Diffusion Process (NDP) performs second best after the gold-standard GPR model. We also included a random search strategy to the baselines. We notice that for Hartmann 3d the random strategy outperforms the NP approach.
>
> > NDP provides an interesting idea to model the joint distribution of a given function space in Section 5.3. However, it is confusing to me since I think it is nothing more than changing the delta distribution in equations (2) and (3). Why do you separate those contents from Section 3?
>
> You are correct, the forward and the backward pass of the NDP for the joint are only different in the delta distribution. During the writing process, we experimented with explaining the full joint model directly in section 3. However, after receiving feedback we found that readers tended to have an easier time understanding our model (and our contributions) when we explained the two approaches in a sequential fashion. It is also worth noting that while eqs. (2) and (3) only change marginally, one would also have to explain a larger network architecture (see Figure 7 in the supplementary) and a more difficult variant of proposition 2 for both $\epsilon_{\theta}^x$ and $\epsilon_{\theta}^y$.
>
> We thank the reviewer for their comments and hope that the strong performance of our NDP model on the requested benchmarks, and the additional clarifications, could lead to an increase in their score.

---

### Author Response · Authors · 2022-11-11
**Changes to revised manuscript based on reviews**

We thank all of the reviewers for their detailed comments. Your feedback has improved the revised manuscript for which we are very grateful.

**Main changes in the revised manuscript:**
- We added Ackley and Rastrigin optimisation problems to the Bayesian optimisation experiment (Figure 6) [Reviewer sTEa and d5b3].
- We clarified the limitations of NDPs by including a limitations section (page 9) describing (1) the effect of the number of diffusion steps on the sampling speed and quality; (2) the lower performance of sampling outside of the training range, as is common with many neural network approaches [1] [Reviewer 6nma and YxyB].
- Added a brief description of spectral kernels and attentive latent NPs to the background section [Reviewer 6nma and YxyB].
- Added a mathematical description of our bi-dimensional attention block to Section 4.2 [Reviewer YxyB].

**References:**

[1] Gordon et al., Convolutional conditional neural processes. ICLR, 2019.

---

### Decision · Program_Chairs · 2023-01-20

**Decision:**

Reject

**Justification For Why Not Higher Score:**

Better justification for the consistency of the "process" and better experimentation on real world problem.N/A

**Justification For Why Not Lower Score:**

N/A

**Metareview: Summary, Strengths And Weaknesses:**

The paper propose to extend diffusion models to define distributions over functions. The paper works on pairs of input , function output , where a score function is learned to denoise noisy output of the function given the position of the input.

A special care is added to ensure permutation invariance w.rt to input /output pairs, and invariance to dimensionality when learning the score function. A new attention is devised for this purpose. The proposed method  is then evaluated on synthetic datasets for hyperparameter marginalisation, non-Gaussian posteriors and baysian  optimisation.

The paper was discussed at length between authors , reviewers and AC.

Reviewers thought that the idea is promising nevertheless it has few issues on fundamental level and on the applications level proposed in the submission.

* consistency of the construction needs to be revised and explained in which sense it is meant in the paper. Especially regarding the backward process as pointed out by reviewer 6nma and acknowledged by the authors.
* The lack of real world data experiments was also pointed out as limiting for the impact of the work by all reviewers.




**Summary Of Ac-Reviewer Meeting:**

The paper was discussed in email threads/ call  between AC and reviewers and  the reviewers were  encouraged to engage in discussion with the authors  to clarify their concerns. The main pain points are summarized above.